# A Perspective on the Development of TGF-β Inhibitors for Cancer Treatment

**DOI:** 10.3390/biom9110743

**Published:** 2019-11-17

**Authors:** Linh Khanh Huynh, Christopher John Hipolito, Peter ten Dijke

**Affiliations:** 1Laboratory of Experimental Pathology, Graduate School of Comprehensive Human Sciences, University of Tsukuba, 1-1-1 Tennodai, Tsukuba, Ibaraki 305-8575, Japan; s1826071@s.tsukuba.ac.jp; 2Cancer Signaling, Faculty of Medicine, University of Tsukuba, 1-1-1 Tennodai, Tsukuba, Ibaraki 305-8575, Japan; hipolito@md.tsukuba.ac.jp; 3Peptide Core Facility, Transborder Medical Research Center, University of Tsukuba, 1-1-1 Tennodai, Tsukuba, Ibaraki 305-8575, Japan; 4Oncode Institute and Cell Chemical Biology, Leiden University Medical Center, 2300 RC Leiden, The Netherlands

**Keywords:** cancer therapy, epithelial-to-mesenchymal transition, immune evasion, signaling, SMAD, TGF-β, tumor microenvironment

## Abstract

Transforming growth factor (TGF)-β is a secreted multifunctional cytokine that signals via plasma membrane TGF-β type I and type II receptors and intercellular SMAD transcriptional effectors. Aberrant inter- and intracellular TGF-β signaling can contribute to cancer progression. In normal cells and early stages of cancer, TGF-β can stimulate epithelial growth arrest and elicit a tumor suppressor function. However, in late stages of cancer, when the cytostatic effects of TGF-β in cancer cells are blocked, TGF-β signaling can act as tumor promoter by its ability to stimulate epithelial-to-mesenchymal transition of cancer cells, by stimulating angiogenesis, and by promoting evasion of immune responses. In this review, we will discuss the rationale and challenges of targeting TGF-β signaling in cancer and summarize the clinical status of TGF-β signaling inhibitors that interfere with TGF−β bioavailability, TGF-β/receptor interaction, or TGF-β receptor kinase function. Moreover, we will discuss targeting of TGF-β signaling modulators and downstream effectors as well as alternative approaches by using promising technologies that may lead to entirely new classes of drugs.

## 1. Introduction

Transforming growth factor β (TGF-β) is part of a larger superfamily of secreted dimeric multifunctional proteins that also includes activins and bone morphogenetic proteins. It plays an important role in embryogenesis and in maintaining tissue homeostasis in multicellular organisms [1,2]. TGF-β elicits highly context-dependent effects on many different cell types [1,3]. Hence, anomalous TGF-β signaling can result in numerous diseases, including cancer. TGF-β signaling can have a dual role in cancer [4]. In normal cells and early stages of cancer progression, TGF-β signaling has tumor suppressor functions, including cell-cycle arrest and apoptosis. However, in later stages of cancer progression, cancer cells can become resistant to TGF-β signaling’s cytostatic effects, but remain responsive. TGF-β signaling can then contribute to malignant progression by promoting epithelial-to-mesenchymal transition (EMT) and thereby promote cancer cell invasion, metastasis, and chemoresistance [5]. Moreover, malignant cancer cells and stromal cells within the tumor vicinity frequently secrete high amounts of TGF-β. This process also promotes tumorigenesis by creating a favorable microenvironment through stimulation of angiogenesis and immune evasion [1,3]. TGF-β’s dual function in cancer and its pleiotropic activities make development of an effective anticancer therapy without unwanted side effects a challenge [5,6]. Here, we present a brief overview of TGF-β signaling, its role in cancer progression, and recent clinical advances and setbacks in targeting aberrant TGF-β signaling. Moreover, the targeting of deregulated TGF-β signaling modulators as well as downstream effectors will be discussed. Finally, we provide a perspective on new technologies for future targeting of TGF-β signaling by interfering with intracellular protein–protein interactions and harnessing the proteasomal machinery to degrade intracellular proteins.

## 2. TGF-β Signaling Pathway

The TGF-β signaling pathway can be activated through the interaction of TGF-β ligand with its cognate type I and type II single-pass transmembrane receptors (i.e., TβRI and TβRII, respectively) that are endowed with intrinsic serine/threonine kinase activity [7]. Three TGF-β isoforms have been identified, TGF-β1, 2, and 3, which share 70% sequence identity, bind the same TGF-β type I and type II receptor complex and activate the same downstream intracellular signaling pathways [8]. The germ line knockout of individual isoforms results in different phenotypes suggesting that it may be attributed to their differential expression patterns [5,8,9]. The TGF-β cytokines are secreted as inactive latent complexes, in which the precursor amino-terminal portion is wrapped around the mature, bioactive carboxy-terminal portion. Latent TGF-β cytokines can be activated by proteases and mechanical force/integrin-dependent processes [10]. Activated TGF-β cytokines initially engage with TGF-β co-receptors, such as betaglycan (also termed TβRIII). These auxiliary receptors are abundantly present on the cell surface, but bind TGF-β with weaker affinity than TβRI and TβRII signaling receptors [11]. Thereafter, TGF-β is presented and bound to TβRII, which subsequently recruits and phosphorylates TβRI at serine and threonine residues located in the so-called glycine-serine (GS) juxtamembrane domain [5]. Upon receptor transphosphorylation, the extracellular signal is successfully transduced across the plasma membrane. The activated TβRI initiates intracellular signaling by phosphorylating downstream effector proteins called SMADs. The activated TβRI phosphorylates receptor-regulated (R)-SMAD2 and SMAD3 at two C-terminal serine residues. The phosphorylated R-SMADs dissociate from the receptor complex, assemble into heteromeric complexes with common mediator (Co)-SMAD4. The SMAD complexes then translocate to the nucleus, where they regulate gene expression [12]. Whereas SMAD3 and SMAD4 can bind directly to DNA, SMAD2 does not [13]. SMAD2 contains an extra exon not present in SMAD3. This extra exon codes for a peptide insert located in close proximity to a **β**-hairpin motif, which, in SMAD3, is responsible for direct contact of with DNA [14]. As a result, this peptide insert prevents SMAD2 from directly binding to DNA. Typical TGF-β signaling target genes are the genes encoding extracellular matrix protein plasminogen activator inhibitor-1 (PAI-1) [13] and SMAD7, which acts as an inhibitor of TGF-β/SMAD signaling [15]. Promoters for both genes have so-called SMAD-binding elements (SBEs) consisting of repeating 5′-CAGA-3′ motifs. SMAD7 antagonizes TGF-β signaling by recruiting E3 ubiquitin ligase SMURF2 to activated TβRI, which targets this receptor for proteasomal and lysosomal degradation [16]. SMAD7 controls the duration and intensity of TGF-β-induced biological responses [15,17] and its induction is a means for crosstalk of other pathways with the TGF-β signaling pathway. The ubiquitination of TβRI can be reversed by USP4/15 deubiquitinating enzymes [18] (Figure 1).

## 3. TGF-β Signaling in Cancer Progression

### 3.1. Biphasic Role in Cancer Progression

In the early malignant stage, TGF-β secreted by tumor or stromal cells plays a tumor suppressor role by inducing cell-cycle arrest and promoting apoptosis [19]. TGF-β causes late G1 cycle arrest by regulating the expression of cyclin-dependent kinases (CDK) inhibitors, p21 and c-Myc, to inhibit cell cycle progression [20,21]. In addition, TGF-β can induce apoptosis by upregulating apoptotic regulators such as Bcl-2-like protein 11 (BIM), BCL-2 interacting killer (BIK), and death associated protein kinase (DAPK) [22,23,24]. Cancer cells often bypass the TGF-β-induced cytostatic effects and apoptosis by mutating key components of the TGF-β pathway [25]. Colorectal cancer-derived TβRII originating from a group of patients with mutation in the genes encoding mismatch repair proteins was often found to be mutated [26,27,28]. High rates of deletion and/or mutation of SMAD4 are present in pancreatic cancers and are also reported in lower frequency in colorectal cancer (CRC) and hepatocellular carcinoma (HCC) [29,30]. Inactivating mutations in the SMAD2 gene have been indicated in CRC, HCC, and lung cancer.

In pre-malignant cancer progression, the function of TGF-β signaling changes from a tumor suppressor to a tumor promoter. In late stage cancer, tumors such as melanomas, gliomas, and breast cancer have maintained a functional TGF-β receptor/SMAD pathway, but are resistant to the tumor suppressive effects of TGF-β due to loss of function mutation in tumor suppressor genes and/or acquiring oncogenic mutations, such as in the PI3K/AKT, RAS/MAPK pathway components [31]. In this case, cancer cells remain responsive to TGF-β signaling, which then can promote EMT, invasion, and metastasis [32] (Figure 2).

### 3.2. TGF-β and EMT 

Upon epithelial-to-mesenchymal transition (EMT), epithelial cells repress their epithelial morphology and gain mesenchymal characteristics, elongate, and acquire motile and invasive properties during the course of development, wound healing, and pathological processes such as fibrosis and cancer [33,34]. Several hallmarks of EMT have been identified, including increased expression of mesenchymal adhesion proteins (N-cadherin, vimentin), and decreased expression of epithelial adhesion proteins (E-cadherin, ZO1, desmoplakin) [33,34,35] (Figure 3A).

TGF-β plays an important role in altering early epithelial cancer cells to invasive metastasis cancer cells by promotion of EMT. TGF-β can activate transcriptional factors such as SNAIL1/2, ZEB1/2 and TWIST1/2 that mediate EMT in these tumors by in a SMAD-dependent manner [5]. In the section below, we provide some examples regarding these EMT transcription factors. A study by Cano et al. showed that zinc finger transcription factor SNAIL is a strong repressor of *E-cadherin* expression [36]. By reducing expression of *SNAIL* and *SLUG*, resistant ovarian cancer cells were resensitized to the chemotherapy drug cisplatin and the mesenchymal phenotype was largely reversed [37,38]. Interestingly, another study indicated that inhibition of SNAIL-induced EMT could suppress both tumor metastasis and immunosuppression in cancer patients. These results implied that targeting SNAIL may be useful for treatment of cancer in multiple ways, including restoration of immunocompetence in patients [39].

TWIST, a basic helix–loop–helix transcription factor and master regulator of embryonic morphogenesis, induces EMT in vitro [40]. A study showed gene silencing of TWIST by small interfering RNA (siRNA) in a murine model of metastatic breast cancer can reduce metastasis [41]. Moreover, TWIST expression correlates with invasive lobular breast cancer carcinoma histology in humans. These results establish a mechanistic link between TWIST, EMT, and tumor metastasis. The potential of TβRII receptor inhibitors and TWIST to relieve metastasis in murine 4T1 mammary carcinoma models indicates the TWIST as a promising downstream target and for combination therapies of these inhibitors [7].

ZEB family transcription factors also have been well documented concerning their role in cancer progression [42]. Both ZEB1 (dEF1) and ZEB2 (SIP1) suppress E-cadherin transcription by binding to the conserved E2 boxes in the *E-cadherin* promoter [43]. A study by Eger et al. demonstrated that downregulation of ZEB1 by RNA interference was sufficient to suppress E-cadherin expression and restore cell adhesion in breast cancer cells [44]. In addition, upregulation of ZEB1 was observed in invasive ductal and lobular breast cancer [45]. Another study also indicated that ZEB1 and SNAIL downregulate E-cadherin expression in cyclooxygenase-2-dependence in non-small cell lung cancer (NSCLC) [46]. Moreover, ZEB1 inducing the loss of basement membrane indicates metastasis and poor survival in colorectal cancer [47] (Figure 3B).

## 4. The Role of TGF-β in Tumor Microenvironment

TGF-β is also responsible for regulating stroma cells in the tumor microenvironment (TME). The TME consists of cancer-associated fibroblasts (CAFs), myofibroblasts, extracellular matrix (ECM), immune/inflammatory cells, blood, and vascular networks [5]. Cancer-associated fibroblasts (CAFs) are present in a large number in the TME and are the main producer of TGF-β. The studies by Calon et al. showed that a group of colorectal cancer patients exhibiting high TGF-β pathway activity in CAFs are prone to metastasis and poor-prognosis [48,49]. CAFs produce interleukin (IL)-11, an inducer of TGF-β, which can prolong the survival of cancer cells by activating the signal transducer and activator of transcription (STAT) 1 pathway [49]. Moreover, TGF-β can differentiate stromal mesenchymal stem cell (MSCs) into myofibroblasts that produce extracellular matrix and growth factors to stimulate tumor growth [50,51,52] (Figure 4).

TGF-β is a potent immunosuppressive cytokine with pleiotropic effects on most immune cells including dendritic cells, macrophages, natural killer cells, CD4+, CD8+ cells [53,54]. TGF-β can also stimulate the differentiation of immune-suppressive regulatory T (Treg) cells [53]. TGF-β acts on cytotoxic T lymphocytes (CTLs) by downregulation of five cytotoxic genes (perforin, granzyme A/B, Fas ligand, and interferon γ) responsible for CTL-mediated tumor cytotoxicity [55]. TGF-β signaling in the TME has been associated with poor prognosis. The TGF-β secreted by cells in the TME can suppress immune response leading to tumor progression [56]. Several studies demonstrate that interrupting TGF-β signaling can enhance antitumor immunity. For instance, T-cell-specific blockade of TGF-β signaling can enhance immune response to eradicate tumor in mice challenged with live tumor cells [57]. In the B cell acute lymphoblastic leukemia (B-ALL), TGF-β secreted by cancer cells can inhibit natural killer (NK) cells, and thereby, tumor cells can escape immune detection [58]. Therefore, a reasonable strategy for boosting immune response would be the inhibition of TGF-β signaling in B-ALL, which can restore NK cells function. In breast cancer mouse models, inhibition of TGF-β can inhibit IL-17 expression by CD8+ T cells, which results in decreased tumor progression [59]. Radiation therapy combined with TGF-β signaling inhibitors can improve the therapeutic effect by reducing immunosuppressive function of TGF-β and by stimulating CD8+ cells cytotoxic response to tumor cells [60].

In recent years, immune evasion has become an important focus of research [54]. Studies conducted by Mariathasan et al. [61] and Tauriello et al. [62] identified that the activation of TGF-β signaling in the CAFs contributes to T cell exclusion, which results in poor response to immune checkpoint PD-1/PD-L1 blockade mediated by atezolizumab. Notably, the lack of response to atezolizumab was associated with a transcriptional signature of TGF-β signaling in fibroblasts. Moreover, they provide preclinical evidence in mouse models indicating that the treatment of TGF-β inhibitor combined with atezolizumab can facilitate CD8+ T cell penetration and tumor regression while the treatment with atezolizumab or TGF-β inhibitor alone is ineffective [61,63]. Another promising application is that CAR-T cell therapy can be engineered to convert TGF-β from an immunosuppressive cytokine to a strong stimulator of T cells [64]. These studies can pave the way for broader application of immunotherapy in cancer patients.

## 5. Inhibitors that Target TGF-β or TGF-β Receptor Function

There are several TGF-β signaling inhibitors such as neutralizing antibodies, ligand traps, and receptor kinase inhibitors that have been tested in (pre)clinical trials [5,6] (Figure 5). This section reviews current TGF-β inhibitors, their clinical status, their advantages, and remaining drawbacks in cancer treatment.

### 5.1. Antibodies That Target TGF-β Activation and Ligand–Receptor Interaction

Antibodies are used to block the activation of latent TGF-β and to interfere with TGF-β ligand binding to its cognate receptor. Both steps are critical for TGF-β to elicit its pro-tumorigenic responses. Integrin αvβ6 has been demonstrated to have a key role in activation of latent TGF-β [65]. Studies have shown that a human monoclonal antibody, 264RAD, targeting αvβ6 integrin can reduce tumor growth and metastasis in vivo [66]. Targeting αvβ6 with 264RAD antibody alone or in combination with trastuzumab, an antibody neutralizing Human Epidermal Growth Factor Receptor 2, can be beneficial for patients with high-risk and trastuzumab-resistant breast cancer [67].

Fresolimumab (GC1008), a human monoclonal antibody neutralizing TGF-β1, TGF-β2 was tested in a Phase I trial composed of 28 patients with malignant melanoma and one patient with renal cell carcinoma. Seven patients showed partial response, four patients developed reversible cutaneous keratoacanthomas and squamous-cell carcinomas and one patient acquired hyperkeratosis as unwanted side effects [68].

LY3022859, an anti-TβRII monoclonal antibody that inhibits receptor-mediated TGF-β signaling activation, was tested in a Phase I clinical trial composed of 14 patients with advanced solid tumors. The maximum dose tolerance was not determined. Dose escalation beyond 25 mg/dose was deemed unsafe due to negative symptoms (uncontrolled cytokines release). Unfortunately, a safe dose at which the drug can show effectiveness could not be determined [69].

### 5.2. Ligand Traps That Sequester Ligands from Receptor Binding 

Soluble TβRII (sTβRII)-Fc and soluble betaglycan (sBetaglycan)-Fc were constructed as Fc fusion proteins, in which immunoglobulin fragment crystallizable (Fc) was fused to the extracellular domain of TβRII and betaglycan, respectively. In a similar manner to the neutralizing antibody, targeting TGF-β by sTβRII-Fc and sBetaglycan-Fc could also reduce tumor growth and metastasis in preclinical models. Muraoka et al. showed that TβRII-Fc-induced apoptosis in primary tumors and reduced tumor cell motility, intravasation, and lung metastases. sTβRII-Fc also inhibited metastases from transplanted 4T1 and EMT-6 mammary tumors in mice [70]. A study showed that injection of sBetaglycan-Fc by peritumoral (50 μg/tumor, twice a week) or intraperitoneal (100 μg/animal, every alternate day) into human breast carcinoma MDA-MB-231 xenografts could inhibit the tumor growth and lung metastasis [71].

Qin et al. showed that they successfully developed a trivalent TGF-β receptor trap, RER, which consisted of a single extracellular domain of betaglycan flanked by two extracellular domains of TβRII [72]. The ligand trap was shown to effectively block the interaction between TGF-β and TβRII, which resulted in inhibition of TGF-β signaling. Moreover, its anti-TGF-β activities were found to be equal or better than that of an anti-TGF-β antibody and a small molecule TβRI inhibitor in various prostate cancer cell lines. RER was shown to inhibit early-stage tumorigenesis and tumor cell invasion in murine Pten-deficient prostate glands. A recent study also showed RER can disrupt chemotherapeutics-induced TGF-β signaling and has antitumor activity in gynecologic cancers [73]. These findings suggest that ligand traps based on TβRs could be useful to antagonize TGF-β signaling in various types of cancer.

### 5.3. Engineered Mutant TGF-β Ligands 

The availability of structural information on TGF-β family proteins provides avenues for engineering these proteins with improved activity either as research tools or novel therapeutic agents. It is possible to alter TGF-β ligands from the activator to the inhibitors of TGF-β signaling with some modification including the deletion of an α-helix and replacement with a flexible loop [74]. It will of interest to examine these antagonists in preclinical cancer models.

### 5.4. Small Molecules

Besides blocking TGF-β ligand-receptor interaction by large molecules, several small molecule drugs have been developed to inhibit receptors’ kinase activity. Like other kinase inhibitor development, these inhibitors are designed to bind to the ATP-binding domain of TβR kinases.

The TβRI kinase inhibitor galunisertib has been tested in Phase II clinical trial for patients with unresectable pancreatic cancer and advanced hepatocellular carcinoma (HCC). Treatment of pancreatic cancer patients with galunisertib in combination with gemcitabine (104 patients) showed improved overall survival (10.9 vs. 7.2 months) compared to the group treated with gemcitabine and placebo (52 patients) (NCT01373164) [75,76]. In another Phase II study, 40 patients with HCC who had progressed on or were ineligible to receive sorafinib were treated with intermittent dosing of galunisertib with 14 days on/off. The result indicated HCC patients with normal alpha-fetoprotein and with TGF-β1 reduction showed improvement in overall survival compared to patients with non-TGFβ1 reduction (NCT01246986) [76]. In preclinical models, toxicity of galunisertib demonstrated cardiac toxicity [77]. This side effect was considered as a consequence of on-target TβRI inhibition rather than off-target effect. This study highlighted the challenges in therapeutically targeting the TβRI [77]. The success of moving galunisertib forward was largely based on the development of an intermittent dosing schedule by predictive pharmacology, pharmacodynamic markers, and preclinical toxicology models [5,78,79]. The optimal effective dosing schedule that can induce antitumor activity, but has no cardiac toxicity, is iterative cycles (28 days for 1 cycle) comprised of 2 weeks treatment with galunisertib followed by 2 weeks without the drug. This protocol was applied to patients with glioma and no serious cardiac toxicity was observed [80,81]. Patients with isocitrate dehydrogenase (IDH) 1 mutation tended to respond to galunisertib. However, the phase II study failed to demonstrate the overwhelming effect of galunisertib as monotherapy or combined with chemotherapy lomustine in patients with recurrent glioblastoma. In part, this may also be caused that the patient selection was not done properly [5].

Taken together, the TGF-β targeting agents have shown promise in pre-clinical models, but the results in clinical trials are modest. The complexity and dual function of TGF-β signaling make it difficult to target without side effects to patients. Some approaches have been tried to improve the outcome of TGF-β inhibitor treatment. A carefully prepared scheduling of intermittent dosing of TGF-β inhibitors, patient selection, and combination therapy are important considerations for enhancing the efficiency of targeting TGF-β signaling [5].

### 5.5. Future Perspective of TGF-β Signaling Inhibitors

Most of current TGF-β signaling inhibitors aim to directly inhibit either TGF-β receptors’ kinase activity or TGF-β cytokine function. However, these strategies will inhibit all functions of TGF-β family signaling. Instead of directly targeting TGF-β and its receptors, there are other genetic approaches for selective inhibition or more indirect ways to control the TGF-β signaling pathway. Here, we discuss alternative genetic approaches for targeting, or (potential) interference by targeting modulators or downstream effectors of TGF-β signaling.

## 6. Targeting TGF-β Signaling Effectors and Modulators

### 6.1. Targeting specific TGF-β Signaling Components by Using Anti-Sense Oligonucleotides 

Antisense oligonucleotides (AONs) are short oligonucleotides designed for silencing target gene expression. Due to the instability of oligonucleotides, nanoparticles and chemically modified AONs have been developed to improve the delivery and stability of oligonucleotides to target cells or tissues [83,84]. AONs have been developed that target selectively TGF-β signaling components (Figure 6A).

TGF-β2 is an established target for glioma therapy. AP12009 (trabedersen) is an 18-mer AON that targets TGF-β2 expression in glioma, pancreatic carcinoma, and malignant melanoma. AP12009 has undergone or is currently use in multiple Phase 3 clinical trials against astrocytoma (NCT00761280). Unlike TGF-β1 or TGF-β2, TGF-β3 has not received as much attention as a therapeutic target. Seystahl et al. focused on TGF-β3 and used phosphorothioate locked nucleic acids (LNA) to target TGF-β3 in glioblastoma [85]. In addition, AONs were designed to specifically induce exon skipping of mouse TβRI receptor transcripts. AON-induced exon skipping of TβRI lead to specific downregulation of full-length receptor transcripts in vitro in different cell types, which reduces TGF-β signaling activity [86,87].

### 6.2. Targeting miRNA, lncRNA

Besides main mediators directly affecting TGF-β pathway, there are other crucial processes indirectly regulating TGF-β signaling responses. Micro (mi)RNA, long non-coding (lnc)RNA, and deubiquitinating enzymes (DUBs) are well-documented regulators of TGF-β signaling and cancer progression, including metastasis. Hence, targeting these modulators and effectors of TGF-β signaling can be an alternative approach for therapeutic development.

Many studies have shown the involvement of miRNA and lncRNAs as important modulators of TGF-β signaling and EMT (Figure 3). For instance, a study showed that overexpression of miR-10b induced TGF-β-driven EMT in breast cancer. Furthermore, silencing miR-10b with antagomirs, a class of chemically modified anti-miRNA oligonucleotides, can reduce lung metastases by inactivating its target gene *HOXD10* both in vitro and in vivo [88]. Administration of miR-10b antagomirs in mice bearing highly metastatic cells does not reduce primary tumor growth, but notably suppresses lung metastases in a sequence-specific manner. The miR-10b antagomir, which is well tolerated by normal animals, appears to be a promising candidate for the development of new anti-metastatic therapies.

A few studies also reported the important role of lncRNAs in these processes. For instance, a study showed lncRNA activated by TGF-β (lncRNA-ATB) induces EMT and invasion by inhibiting negative feedback regulation of TGF-β signaling. In another study, reduced lncRNA H19 expression in hepatocarcinogenesis tissues from patients is associated with the epithelial TGF-β gene signature while increased H19 expression promoted tumor metastasis in Hep3B cells. In metastatic breast cancer mouse models, lncRNA H19 mediates the plasticity of EMT and mesenchymal to endothelial transition (MET) by binding to miR200b/c and let 7-b. LncRNA H19 also stimulates EMT by interacting with SLUG or EZH2 [89]. LncRNA HOTAIR acts as important player in EMT activation by mediating the physical interaction between SNAIL and EZH2 [90,91]. These studies suggest that lncRNAs and can be effectors of TGF-β signaling and potential targets for anti-metastatic therapies.

Many studies show the improved efficiency of chemotherapy by combination with the delivery of siRNA targeting EMT-TFs [92]. Robert et al. showed that cisplatin combined with the delivery of TWIST siRNA by nanoparticles results in lowering epithelial ovarian tumor burden in mice than treatment with cisplatin alone [93]. In addition, other delivery systems have been used to enhance efficiency of siRNA delivery in vivo such as polypeptide micelle nanoparticles and recombinant adeno-associated viruses (rAAVs). In principle, using siRNA for direct targeting EMT-TFs appears promising but the delivery system remains to be improved [92] (Figure 6A).

### 6.3. Targeting of Deubiquitinating Enzymes (DUBs) 

Deubiquitination has been reported to play main role in regulating TGF-β signaling such as protecting the receptor R-SMADs and Co-SMADs from degradation, which maintains TGF-β signaling [94]. Thus, targeting DUBs that are highly expressed/active in advanced cancer and that mediate TGF-β-induced pro-oncogenic response can be a potential therapeutic strategy (Figure 6B). For example, USP4, USP15, and USP11 were found to deubiquitinate TGF-β type I receptors, and thereby stabilize the receptors [94]. USP4 can interact and stabilize TGF-β type I, which promotes invasion of breast cancer [95] (Figure 1). High USP15 expression is correlated with enhanced pSMAD2 expression in tissue samples of glioblastoma patients. Moreover, inhibition of USP15 leads to the decrease in TGF-β type I receptor levels and phosphorylated SMAD2 concentrations in these cells thus inhibiting TGF-β/SMAD2 signaling, and glioma progression [96]. Based on these finding, specific pharmacological inhibitors for USP4 and USP15 may therapeutic potential in cancer treatment.

One of the main challenges in developing specific inhibitors for DUBs is that the binding pockets of many DUBs are not optimal for small molecule binding. However, the first selective DUB inhibitors have been generated [97]. Furthermore, DUBs have multiple substrates. Hence, inhibiting protease activity of DUBs by small molecules can also lead to unwanted side effects. One possible strategy to overcome these limitations could be modulating DUB activity by targeting protein–protein interactions [96,97,98].

### 6.4. Interfering with Intracellular Protein-Protein Interactions

To determine whether disrupting SMAD protein–protein interaction would selectively inhibit TGF-β response, Cui et al. demonstrated that SMAD-interacting peptide aptamers from FOXH1, LEF1 and CBP could be used to selectively inhibit TGF-β-induced gene expression. The result suggested that these peptide aptamers might be useful for disrupting subsets of TGF-β responses [99]. Another study provides the evidence that the cell-penetrating peptide sorting nexin 9 (SNX9) can specifically target phosphorylated SMAD3 and inhibit profibrotic TGF-β signaling in vitro and in vivo [100]. The peptide was developed by conjugation of SNX9, a member of the PX/BAR subfamily of intracellular trafficking protein and cell-penetrating HIV-TAT protein, a key HIV transactivator of transcription [101]. Furthermore, targeting intracellular SMAD signaling appears to be a promising strategy to selectively inhibit TGF-β tumor promoter function. Particularly, many studies in murine models showed that targeting SMAD3 could reduce bone metastasis in breast cancer [32,102,103,104]. From clinical perspectives, targeting transcription factors such as SMADs, which lack intrinsic enzymatic activity, is challenging since protein–protein or protein–DNA interaction surfaces often lack deep binding pockets. However, the recently approved small molecule Bcl2 inhibitor Venetoclax demonstrates that this approach is reasonable [34,35]. In addition, there are a number of emerging technologies that make this approach of inhibiting non-enzymatic intracellular protein function more promising.

## 7. Cyclic Peptides and PROTAC as Potential Technologies for Targeting TGF-β signaling

Cyclic peptides may offer the ideal scaffold for disrupting protein-protein interaction [105,106,107,108]. (Figure 6C). In 2006–2015, nine cyclic peptide drugs have been approved by FDA, three of which are oncology drugs namely lanreotide, romidepsin, and pasireotide [109]. In addition to the currently approved cyclic peptide therapeutics, the majority of which were derived from natural products, the advent of powerful techniques based on rational design and in vitro evolution have enabled the relatively routine discovery of cyclic peptides against targets of interest. However, there are two main challenges remaining in the development of cyclic peptide therapeutics: oral availability and cell permeability [109,110,111]. The strategies are shown to overcome these challenges include the use of cell penetrating peptides [110], stapled peptides [112,113], or incorporation with *N*-methylated amino acids [114]. Based on these proof-of-principle studies, cyclic peptides, especially, those targeting protein–protein interaction are expected to go into clinical use in the near future.

Proteolysis-targeting chimeras (PROTACs) are bifunctional molecules that recruit a target protein to an E3 ubiquitin ligase, which initiates the protein degradation process. In principle, the target specific portion of PROTAC does not need to inhibit the protein’s function since the second molecule would tether the target protein to ubiquitin ligase, which conjugates ubiquitin to the target protein (Figure 6B). The prominent example of this concept is the development of PROTAC molecules for targeting androgen receptors. In 2019, the first oral PROTAC drug (ARV-110, targeting the androgen receptor for degradation) has been reported in phase I clinical trial for patients with metastatic castration-resistant prostate cancer (NCT03888612). This technology offers new opportunity for regulating “undruggable” targets [115].

## 8. Concluding Remarks

TGF-β acts as tumor suppressor in normal tissues and early stage of cancer and acts as a tumor promoter in late stage of cancer. The biphasic role of TGF-β in cancer progression makes it a challenging target to develop therapeutics without unwanted side effects. Notably, TβR kinase inhibitors such as galunisertib have shown therapeutic effects in some cancer patients. The side effects can be mitigated by intermittent dosing schedules and patient selection. Recently, the role of TGF-β as an immunosuppressive cytokine has also been investigated in anticancer therapeutics with some promising results. In addition, targeting of TGF-β signaling effectors, modulators, and its downstream pathway can be considered as alternative approaches to selectively inhibit the pro-TGF-β-induced pro-oncogenic signaling. Thanks to the advanced progress in drug-making technologies, some bottleneck problems of drug development on targeting non-enzymatic intracellular proteins in the past are expected to be untangled in the near future.

## Figures and Tables

**Figure 1 biomolecules-09-00743-f001:**
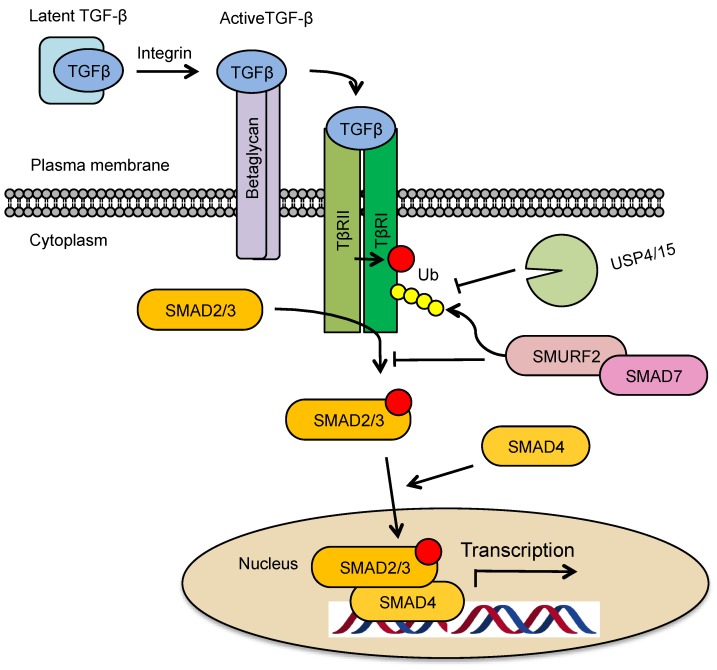
Schematic representation of transforming growth factor (TGF)-β/SMAD signaling. TGF-β is secreted in a latent form, which can be activated via integrin-dependent processes. Activated TGF-β initially engages with co-receptor beta glycan. Thereafter, it is passed on to TβRII that recruits TβRI forming a heteromeric signaling complex. Upon TβR1 phosphorylation and activation by TβRII kinase (phosphorylation indicated with red circle), SMAD2 and SMAD3 are phosphorylated (phosphorylation indicated with red circle) and form complexes with SMAD4. These complexes translocate into nucleus and act as transcription factors to regulate the expression of TGF-β signaling target genes. SMAD7, together with E3 ubiquitin ligase SMURF2, can induce proteasomal/lysosomal degradation of TβRI. This process can be reverted via the action of USP4/15 deubiquitinating enzymes.

**Figure 2 biomolecules-09-00743-f002:**
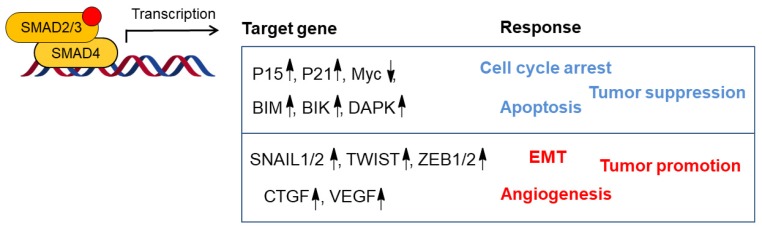
Biphasic role of TGF-β/SMAD signaling in cancer progression. TGF-β has a dual function in cancer as a tumor suppressor processes (in blue) and a tumor promoter processes (in red). In normal cells and early stage cancer cells, TGF-β acts as tumor suppressor by inducing cell cycle arrest and apoptosis. However, in late-stage cancer, increased TGF-β signaling can promote cancer progression including epithelial-to-mesenchymal transition (EMT) and angiogenesis. Cellular responses to TGF-β signaling are mediated via SMAD2/3-SMAD4-dependent regulation of specific target genes.

**Figure 3 biomolecules-09-00743-f003:**
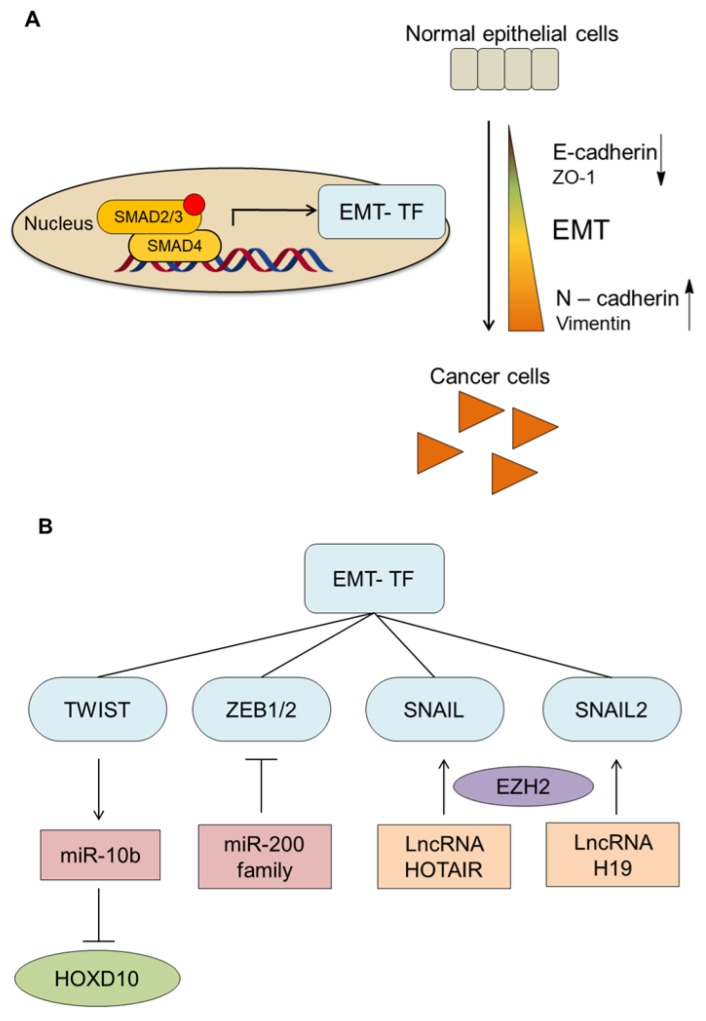
TGF-β signaling mediates EMT. (**A**) TGF-β is an important player in the activation of EMT, which is characterized by downregulation of epithelial markers and upregulation of mesenchymal markers. TGF-β via either SMAD or non-SMAD signaling can enhance the expression of EMT inducing transcription factors (EMT-TF) such as SNAIL1/2, ZEB1/2, and TWIST. (**B**) The involvement of miRNA and lncRNAs as important modulators of TGF-β signaling and EMT: miR-10b induces TGF-β-driven EMT by expression of HOXD10. MiR-200 family can negatively regulate ZEB1/2. LncRNA H19 upregulates EMT by interacting with SLUG or EZH2. LncRNA HOTAIR induces EMT by mediating the physical interaction between SNAIL and EZH2.

**Figure 4 biomolecules-09-00743-f004:**
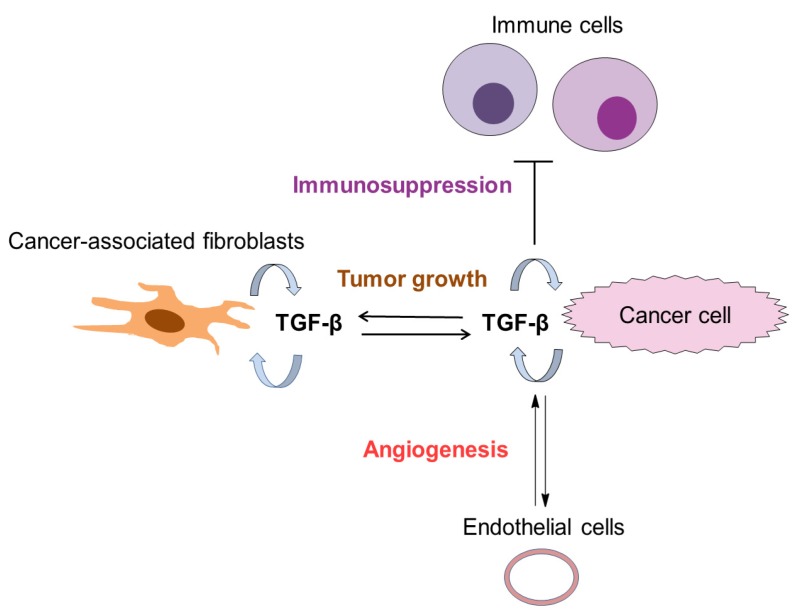
TGF-β signaling and the tumor microenvironment. TGF-β is expressed by cancer and stromal cells including cancer-associated fibroblasts (CAFs). TGF-β can maintain tumor progression by activating CAFs, stimulating immunosuppression, and promoting angiogenesis.

**Figure 5 biomolecules-09-00743-f005:**
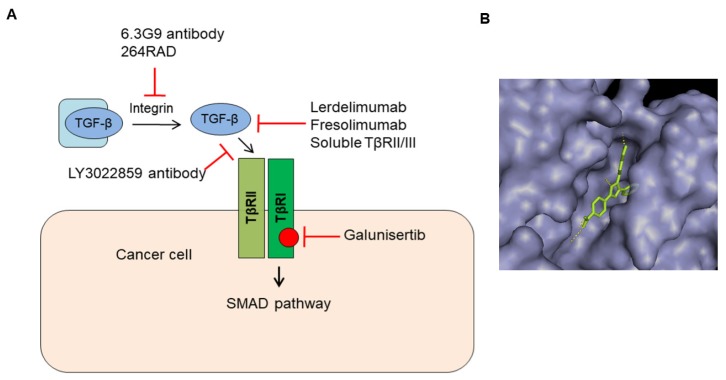
Targeting TGF-β signaling in cancer. (**A**) Various TGFβ signaling inhibitors including neutralizing antibodies, ligand traps, and receptor kinase inhibitors are depicted. (**B**) The structure of TβRI kinase (gray) with its ATP competitive inhibitor: a 5,6-dihydro-4H-pyrrolo[1,2-b]pyrazole analogue (green) (Accession code:1RW8) [82].

**Figure 6 biomolecules-09-00743-f006:**
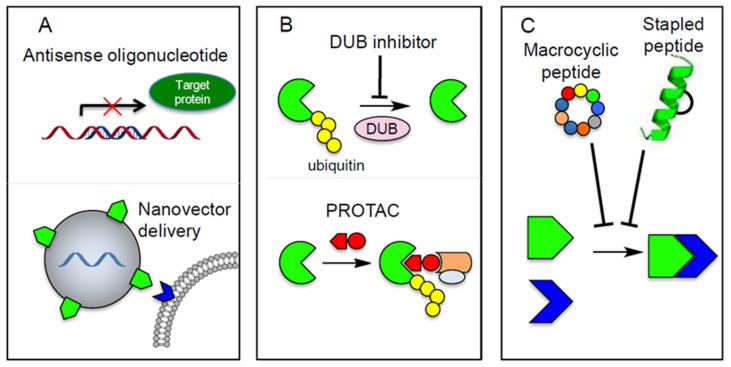
Other possible strategies for targeting downstream TGF-β signaling. (**A**) Targeting TGF-β signaling or its effectors (miRNA, LncRNAs) by antisense oligonucleotides. The stability and specific cell/tissue targeting of the oligonucleotide therapeutic molecules can be enhanced/mediated by nanoparticles. (**B**) Targeting TGF-β signaling inspired by harnessing the proteasomal machinery to degrade intracellular proteins. Deubiquitinating enzyme (DUB) inhibitor can enhance the degradation (or change the activity) of TGF-**β** signaling proteins (e.g., TβRs, SMAD3, EMT-TFs) by inhibiting the deubiquitination process. Likewise, proteolysis-targeting chimeras (PROTACs) can induce the proteasomal-mediated degradation of target proteins (SMAD3, EMT-TFs) by recruiting target protein to specific E3 ligases. (**C**) Interfering protein-protein interaction by cyclic peptides (left) and stapled peptides (right).

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
