# Peer review of "A Perspective on the Development of TGF-β Inhibitors for Cancer Treatment"

_biomolecules, 2019, doi:10.3390/biom9110743_

Round 1

Reviewer 1 Report

This is an excellent and important review of the development of various TGF-β inhibitors that target the ligand, the receptor and receptor kinase activity. Recent advances in the cancer field and the prominence of the involvement of the TGF-β signaling pathway highlight the timeliness and utility of this review by a pioneer corresponding author.  Given the issues surrounding the difficulties and controversies with regard to targeting the TGF-β network for therapeutic gain, the current description of the clinical status of the previous and newer inhibitors is a major contribution to the field. The paper is both comprehensive and very nicely written. Publication is recommended as it.

Author Response

We thank the reviewers positive comments. We have gone carefully through the review again and corrected spelling mistakes.

Reviewer 2 Report

The manuscript by Khanh and colleagues provides a Review regarding the challenges and current status of developing TGF-β inhibitors for cancer treatment.  It is well organized with appropriate references related to the pertinent work in the field.  However, the Review could have a greater impact if the TGF-β superfamily ligands were added to the global overview.  For example, either by adding a summary table or a schematic representation of TGF-β superfamily member(s) signaling cascade, the readers would develop a better understanding of this critical superfamily and some key advancements associated with selective therapeutic promise especially in cancer.  Additional suggestions to strengthen the presentation are offered below for the authors’ consideration: 

There should be consistency in hyphenating the phrase epithelial-to-mesenchymal. On page #6, line 183, “a” should be added between “in” and “large”. On page #9, line 296, the sentence should be rephrased. In Figure 5B, the reference to the structures shown should be further clarified with the designations of green and grey to differentiate them. On page #10, line 332, change “regulate” to “regulating”. On page #12, line 435, “an” should be added between “as” and “immunosuppressive.”

Author Response

We included information that TGFβ is part of larger superfamily that also includes activins and bone morphogenetic proteins. This review is focussed on TGF-beta and not other family members. We therefore do not like to include a table with all family members as it would give the impression that this review will also discuss other family members.

We corrected the grammar as per the second reviewer’s suggestions. On page #7, line 190, “a” was added between “in” and “large”. In Figure 5B, the reference to the structures was clarified with the designations of green and grey to differentiate them. On page #11, line 341, change “regulate” to “regulating”. On page #13, line 446, “an” was added between “as” and “immunosuppressive.”

In addition, other small textual corrections were made

Round 2

Reviewer 2 Report

Excellent revision